# MExMI: Pool-based Active Model Extraction Crossover Membership Inference

**Yaxin Xiao**
The Hong Kong Polytechnic University
20034165r@connect.polyu.hk

**Qingqing Ye**
The Hong Kong Polytechnic University
qqing.ye@polyu.edu.hk

**Haibo Hu**[*]
The Hong Kong Polytechnic University
haibo.hu@polyu.edu.hk

**Huadi Zheng**
The Hong Kong Polytechnic University
huadi.zheng@connect.polyu.hk

**Chengfang Fang**
Huawei International, Singapore
fang.chengfang@huawei.com

**Jie Shi**
Huawei International, Singapore
shi.jie1@huawei.com

## Abstract

With increasing popularity of Machine Learning as a Service (MLaaS), ML models trained from public and proprietary data are deployed in the cloud and deliver prediction services to users. However, as the prediction API becomes a new attack surface, growing concerns have arisen on the confidentiality of ML models. Existing literatures show their vulnerability under model extraction (ME) attacks, while their private training data is vulnerable to another type of attacks, namely, membership inference (MI). In this paper, we show that ME and MI can reinforce each other through a chained and iterative reaction, which can significantly boost ME attack accuracy and improve MI by saving the query cost. As such, we build a framework MExMI for pool-based active model extraction (PAME) to exploit MI through three modules: "MI Pre-Filter", "MI Post-Filter", and "semi-supervised boosting". Experimental results show that MExMI can improve up to $11.14\%$ from the best known PAME attack and reach $94.07\%$ fidelity with only 16k queries. Furthermore, the accuracy, precision and recall of the MI attack in MExMI are on par with state-of-the-art MI attack which needs 150k queries.

## 1 Introduction

Recent advances in machine learning (ML) has significantly shifted the paradigm in all walks of life. Thanks to cloud computing, every business and individual can host black-box ML models in the cloud (e.g., Microsoft Azure ML, Amazon AWS ML, and Google Cloud AI) to provide pay-per-query predictive services, known as Machine Learning as a Service (MLaaS). Obviously, ML models are proprietary assets to the providers who spend great efforts training them [22]. However, by exploiting the correspondence between queries and prediction results from an MLaaS model, an adversary could learn the internals of that victim model to a large, or even full extent. Such attack on model's confidentiality is known as model extraction (ME) [43]. It is a fundamental attack in adversarial machine learning because it enables follow-up attacks such as adversarial samples [19], model evasion [7], and model inversion attacks that infer private information of the training set [16, 48].

State-of-the-art ME attacks can be categorized into direct recovery [23] and active-learning-based ME attacks [6, 34]. Active learning (AL) refers to those semi-supervised training methods that aim to

---

[*]Corresponding author

36th Conference on Neural Information Processing Systems (NeurIPS 2022).

find the most informative training dataset with a limited query budget [17]. Since both AL and ME aim to train a model with as few queries as possible, AL has become increasingly popular in model extraction attacks. Depending on the availability of real-life samples for querying, AL-based ME can be further divided into pool-based active model extraction (PAME) and query-synthesizing-based active model extraction (SAME). The former assumes the presence of samples, from which query samples are iteratively selected using a pool-based or stream-based AL. Classic PAMEs include ActiveThief [34] and Knockoff [33]. The latter does not assume the presence of such samples and obtains them by generative methods [42, 6]. Classic SAMEs include black-box DNNs [35] and PRADA [26].

However, a **critical difference between pool-based AL and pool-based ME** has been long neglected. Since pool-based AL is essentially a training method, the data pool is where training samples are drawn, which means this pool must have the same feature distribution as the training dataset. However, in an ME attack the adversary has no access to the victim model's training dataset or even samples in the same problem domain [10]. Unfortunately, **all previous PAME works ignore this difference and assume the samples in the pool are homogeneous** to those in the training dataset. To demonstrate how non-homogeneous datasets can affect the ME performance, we build a Wide-ResNet-based victim model using the first 25k training image samples of CIFAR10. Then we extract this model using ActiveThief [34], a state-of-the-art PAME, from three adversary data pools: (1) pool $A$ consists of the second 25k training samples of CIFAR10, (2) pool $B$ consists of the middle 25k training samples, and (3) pool $C$ consists of the first 25k training samples. In other words, the victim's training set does not overlap with $A$ (non-homogeneous) but overlaps with half of $B$ (quasi-homogeneous), and completely with $C$ (fully homogeneous). Table 1 shows the fidelity (i.e., similarity to the victim model) of the extracted copy models from $A$, $B$ and $C$. We observe that homogeneous samples contributes more to the success of model extraction than non-homogeneous samples.

In this paper, we address this issue by **identifying homogeneous samples in the data pool** and making full use of them for a "guided" model extraction. We exploit membership inference (MI), an attack that infers the training samples from a given dataset [41], to select them and train a copy model. In turn, the extracted model can enhance the accuracy of membership inference. As such, we propose an iterative extraction framework MExMI where ME and MI reinforce each other through iterations. As such, within limited query budget, the final outcome consists of both a high-fidelity copy model and an accurate set of training samples.

Table 1: Homogeneous v.s. Non-Homogeneous Data Pool for Model Extraction

| The Adversary Data Pool | **A** | **B** | **C** |
|---|---|---|---|
| Training Data Homogeneity | **None** | **Partial** | **Full** |
| Fidelity / % | 90.32 | 91.29 | 92.30 |

There are several challenges in the design of MExMI. First, existing MI attacks do not consider query cost [41, 39], which is essential in ME attacks. We exploit the training data property of the copy model to conduct MI attacks without consuming query budget. Second, state-of-the-art MI attacks impose assumptions that are not available in MExMI or PAME in general. For example, a shadow-model MI requires a labelled dataset drawn from the same distribution as the model's training dataset with the same size [41]. In this paper, we propose a quality metric to evaluate and optimize a shadow model without the need of a large labelled dataset. With this technique, we adapt both shadow-model MI and unsupervised MI [39, 47] to the MExMI framework. Third, existing PAME attacks fail to utilize the training samples purified from the pool. We designed three modules to facilitate PAME to make the most of it. In summary, we make the following contributions:

- We propose a taxonomy (SAME/PAME) on active-learning-based model extraction attacks, and an iterative framework MExMI where PAME and MI reinforce each other.
- We adapt both shadow-model MI and unsupervised MI to MExMI. To boot-strap shadow-model MI, we develop an indicative quality metric of shadow models and design a metric-based shadow-model training algorithm.
- We conduct extensive experiments to compare the effectiveness of MExMI with state-of-the-art PAMEs [34]. MExMI outperforms the latter by $11.14\%$ to $94.07\%$ in terms of ME fidelity, and its MI attack achieves $84.13\%$ precision, on par with the state-of-the-art MI attack [39] ($75.25\%$) that assumes unlimited query budget.

The rest of the paper is organized as follows. Section 2 introduces the background and definitions of ME and MI attacks. Section 3 overviews the MExMI framework and its key components, and

Section 4 introduces adaptive MI algorithms for MExMI. Section 5 presents the evaluation results of MExMI through extensive experiments. Section 6 reviews related works, and Section 7 concludes this work.

## 2 Background and Definition

### 2.1 Victim Models

Our target victim models are multi-class neural network (NN) classifiers, where the input domain is $\mathbb{X} \subseteq \mathbb{R}^d$ and the output is a probability vector of classes $\mathbb{Y} \subseteq \mathbb{R}^K$. Here $d$ denotes dimensionality of the input data, and $K$ denotes the number of output classes. A neural network model makes inference on the output class probabilities $\mathbf{y}$ based on the input $\mathbf{x}$: $\mathbf{y} = [\Pr(Y = 0|\mathbf{x}), \ldots, \Pr(Y = K|\mathbf{x})]$. To train such a model, we assume a supervised learning process.[2] Let $D = (\mathbf{x}_i, \mathbf{y}_i)_{i=1}^n \subseteq \mathbb{X} \times \mathbb{Y}$ denote the victim's training dataset with $n$ labelled samples, where $\mathbf{x}_i$ is the $i$-th training sample, and $\mathbf{y}_i$ is its one-hot format label vector.

### 2.2 Threat Model

We assume a well-trained black-box **victim model** $F$ is deployed in a machine-learning-as-a-service (MLaaS) with a chargeable query interface. For ease of presentation we do not consider the case when the victim employs anti-ME [26] or anti-MI defenses, such as perturbation on the output probability vector [52, 53]. We also assume that the adversary either only has black-box access to the MLaaS model, or knows its architecture (e.g. in AWS Marketplace [1] and Huawei AI Gallery [2]). In addition, the adversary can collect a large number of unannotated public samples to construct an adversary data pool $P$.

### 2.3 Problem Definition

We assume the adversary performs ME attacks against $F$ within a query budget $b$, with the aim to produce a **copy model** $F'$. The main objective of ME is to let $F'$ approximate the victim model $F$ with high degree of resemblance, a.k.a.,**fidelity** [23], which is the proportion of label agreement of two models on an evaluation dataset $D_t$.

## 3 Model Extraction Crossover Membership Inference

In this section, we present our iterative model extraction framework MExMI where ME and MI reinforce each other. An MI attack aims to distinguish those individuals $\hat{D}$ from a population $P$ that exist in the victim model $F$'s **training dataset** [41]. As illustrated in Fig. 1 and pseudocode in Appendix A, the input of MExMI is an adversary data pool $P$ and the access to a black-box victim model $F$, and its outputs are the copy model $F'$ and the inferred training dataset $\hat{D}$. In the first iteration, the adversary chooses $k$ initial seed samples $[\mathbf{x}_1, \ldots, \mathbf{x}_k]_0$ from $P$ without putting them back, where $k$ is the query budget per iteration. These samples are fed to the victim model $F$, which outputs a probability vector $F(\mathbf{x})$. Then an MI attack model $F_{MIA}$ is constructed using the queried dataset $[\{\mathbf{x}_1, F(\mathbf{x}_1)\}, \ldots]_0$ or the copy model (see Section 4). $F_{MIA}$ is used in both MI Post-Filter and MI Pre-Filter modules. The queried dataset is then passed through the MI Post-Filter, which weighs them according to their probability of being a training sample of the victim model. Then the weighted queried dataset is used to train the copy model $F'$, which is then fed to the MI Pre-Filter to refine the adversary data pool for AL sample selection of $k$ queries in the next iteration. The process is repeated until the total query budget $b$ is depleted. Thereafter, without consuming any query budget, $F_{MIA}$ is used to launch MI attacks on $F'$ and the data pool to obtain the final inferred training dataset $\hat{D}$. This dataset will be used in a semi-supervised learning on $F'$ to release the final copy model $F'$.

From Fig. 1, MExMI has three key modules on top of the basic PAME iterative framework [34], namely, MI Pre-Filter, MI Post-Filter and semi-supervised boosting that will be elaborated in rest of this section. As will become clear, they are orthogonal to each other, so they can be turned on or off

---

[2]Throughout this paper, we exclude special training algorithms, e.g., co-training, mutual mean-teaching, and sharpness-aware minimization [46, 49, 15]. Neither the victim nor the adversary uses these algorithms.

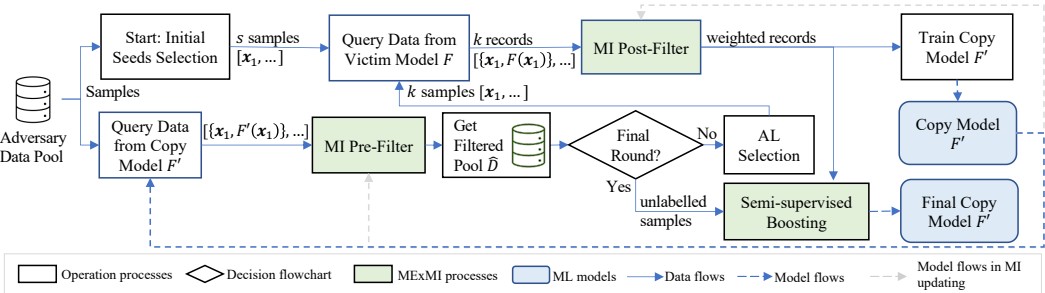

Figure 1: MExMI iterative framework.

independently. For ease of presentation, the construction of the MI attack model $F_{MIA}$, a key issue in the MExMI framework, will be back introduced in Section 4.

### 3.1 MI Pre-Filter

MI Pre-Filter works before AL sample selection. The core idea is to use an MI attack model $F_{MIA}$ to choose from the adversary data pool $P$ only those samples that are highly homogeneous to the victim's training data $D$. Training the copy model with them, both models can thus exhibit strong resemblance. Ideally, $F_{MIA}$ should perform membership inference attack directly on the victim model $F$, which is truly trained from $D$. However, since such training causes extra query budget on $F$, $F_{MIA}$ attacks the copy model $F'$ instead as the latter exhibit similar training data property.

### 3.2 MI Post-Filter

MI Post-Filter works after querying a sample $\mathbf{x}$ from the victim model. The rationale of this filter is that since the victim model returns the probability vectors $F(\mathbf{x})$, the adversary can infer if $\mathbf{x}$ belongs to the training dataset $D$ by an MI attack model $F_{MIA}$. Obviously a negative membership result means this sample may not lead to a high-fidelity copy model $F'$, so its contribution to the training process should be lowered by reducing its weight in the training loss calculation. We use a parameter $\omega$ ($\omega > 1$) to denote the weighted loss ratio of a positive membership sample to a negative one.

### 3.3 Semi-Supervised Boosting

The main idea of this module is that MExMI gain the results of its MI attack— an inferred training dataset $\hat{D}$ of the victim model. Although this dataset is not labelled and there is no more query budget to label them in the end of MExMI, we can still train the copy model $F'$ on this dataset together with the queried set using *semi-supervised learning* algorithms. Note that this module is unique in MExMI as other PAME methods cannot distinguish training samples from others in the data pool. Nonetheless, this module is intended to improve model accuracy only, not fidelity, because semi-supervised learning can divert the copy model's ability to follow the same label distribution as the victim model's training data. Therefore, **MExMI only applies semi-supervised boosting after the final iteration and when higher accuracy is needed**.

## 4 Adaptive Membership Inference

In MExMI an MI attack model $F_{MIA}$ is trained after the initial seeds query. For MI to play a role in MExMI framework in early stage iterations, $F_{MIA}$ must be accurate enough even when there are only few samples. In this section we renovate existing MI algorithms to be adaptive to their training sample sizes and thus suitable for the MExMI framework. We focus on two state-of-the-art black-box MI attack paradigms: (1) shadow-model MI [41, 39], and (2) unsupervised MI [39].

### 4.1 Shadow-Model Membership Inference

The rationale of shadow-model MI is to obtain a shadow model similar to the victim model, so that their output probability vectors for training and non-training samples are also distinguishable in a similar manner. As such, the adversary can build a binary MI classifier from these samples instead of

from those of the victim model. In order to approximate the victim model, a shadow model should (1) draw its training dataset from the same distribution as in the victim model, (2) have roughly the same size of training set as the victim model, and (3) have the same training algorithm. However, in MExMI, neither (1) nor (2) holds because:

**a)** The queried samples may not be drawn from the same distribution as in the victim's training samples.

**b)** The number of queried samples is significantly smaller than that of the victim's training set size, especially in the beginning phase.

As such, the design principle of our adaptive shadow-model MI attack is to work under **a limited number of labelled training samples in a different distribution** from the victim model. To start with, we need to know how to estimate the quality of a shadow model so that we can tell when it is good enough for MExMI. Intuitively, the fidelity of the shadow model against the victim model can serve as the performance indicator, but it is inaccessible from the adversary's side, especially when MExMI just starts. As such, we need an easy-to-access performance metric.

### 4.1.1 Measuring Quality of Shadow-model MI

An MI attack works by distinguishing the output probability vector distribution $\mathbb{Y}$ of the victim model $F$ on training samples $\mathbb{X}$ from non-training samples. A shadow-model MI estimates the above on a shadow model — it distinguishes the output probability vector distribution $\mathbb{Y}^{(s)}$ of shadow models $F_s$ on training samples $\mathbb{X}^{(s)}$ from non-training samples. As such, to improve the attack accuracy, $\mathbb{Y}^{(s)}$ should be as similar as possible to the victim's $\mathbb{Y}$. We measure the similarity by the bias of the expectation values between $\mathbb{Y}^{(s)}$ and $\mathbb{Y}$, denoted by $\mathbf{b}$. For each shadow model that targets at class $j \in [1, \ldots, K]$, $\mathbf{b}_j$ is formally defined as:

$$\mathbf{b}_j = \frac{1}{n_j^{(s)}} \sum_{\mathbf{y}^{(s)} \in \mathbb{Y}_j^{(s)}} \mathbf{y}^{(s)} - \frac{1}{n_j} \sum_{\mathbf{y} \in \mathbb{Y}_j} \mathbf{y}, \tag{1}$$

where $n$ denotes the size of training set, and the superscript $(s)$ denotes the shadow model. The following theorem shows that the gap between the training loss $\Delta l$ is positively correlated to the bias of expectations between $\mathbb{Y}^{(s)}$ and $\mathbb{Y}$. Therefore, it can serve as a quality measurement for $F_s$. We can minimize it to enhance the quality of $\mathbb{Y}^{(s)}$ approximating $\mathbb{Y}$.

**Theorem 1 (Quality measurement for shadow-model MI).** Given a shadow model $F_s$ which has the same model structure and hyper-parameters as the victim model $F$, the gap $\Delta l$ between the training loss of $F_s$ and $F$ is positively correlated to the bias of expectations between $\mathbb{Y}^{(s)}$ and $\mathbb{Y}$, i.e., $\Delta l \propto \mathbf{b}$. See Appendix C for the proof.

### 4.1.2 Metric-based Shadow-model MI

To obtain $\Delta l$, the gap between the training loss of both models, we need them both. However, the victim's training loss is not available to the adversary. Nonetheless, in practice a victim model is valuable for extraction mainly because this model accurately predicts the training data, or equivalently, its training loss is smaller than other models of the same training set size. As such, we can replace $\Delta l$ with the loss of the shadow model, denoted by $l_s$, compensated by its training dataset size $n^{(s)}$. Furthermore, for the sake of comparing various shadow models, only the relative rather than the absolute value of the gap between shadow models and the victim model $\Delta F$ is needed. So we propose the following metric $Q$ as a negative relative value of $\Delta F$. The larger the $Q$, the better the quality of a shadow model. Formally,

$$\frac{1}{\Delta F} \sim Q(l_s, n^{(s)}) = f(l_s) \times (n^{(s)})^a, \tag{2}$$

where $f(\cdot)$ is a non-negative non-linear decreasing function, and $a \in [0, 1]$ scales down the impact of $n^{(s)}$. Note that $Q$ does not require that shadows' training data come from the same distribution as the victim's.

From the above equation, there are two ways to increase the quality metric $Q$: (1) reducing $l_s$ using good training algorithms on shadow models, and (2) increasing $n^{(s)}$ by augmenting the training set. Few-shot learning (FSL), a training paradigm to improve models' accuracy with a limited number

of examples [45], can serve both purposes. For example, we can use FSL approaches, namely data augmentation [29] and transfer learning [21], to train shadow models. This leads to two metric-based shadow-model MIs, which are elaborated in Appendix D.

## 4.2 Unsupervised Membership Inference

Recent works [39, 47] have shown the effectiveness of unsupervised learning on MI attack models. In such models, the feature values are usually the top-$m$ score, loss or entropy of the output probabilities, and the output value serves as the confidence of membership inference — if the value is higher than an adversary-specified threshold $c$, the sample is inferred as in the training dataset and vice versa. To set this threshold, the adversary first gets a set of non-member samples and then query them to get corresponding top-$m$ scores. The top $t$ percentile of these scores can serve as a threshold [39]. To save the query cost, the copy model instead of the victim model should be queried.

# 5 Experiment

In this section, we first conduct experiments to validate the shadow model quality metric $Q$. Then we evaluate the attack performance of four variants of MExMI — Pre-Filter only, Post-Filter only, MExMI without semi-supervised boosting, and MExMI — against state-of-the-art ME and MI attacks. The codes are available at `https://github.com/mexmi/mexmi-project`.

## 5.1 Experiment Setup

**Datasets**.  We perform PAME attacks on two image datasets, namely CIFAR10 [28] and Street View House Number (SVHN) [32], and a text dataset AG'S NEWS which contains corpus of AG's news articles [11] (see Appendix E.1 for details).

**Victim Model**.  For the image classification task, we use Wide-ResNet-28-10 [49] trained on CIFAR10 as the victim model with an accuracy of $96.10\%$. We also use a cloud MLaaS, ModelArts [3], to train an online victim model on SVHN that leads to $94.30\%$ accuracy. In the text classification task, we use DPCNN [25] as the victim model which achieves an accuracy of $89.88\%$.

**Adversarial data pool**.  In the default CIFAR10 experiments, the pool consists of 50k training samples and 100k from the ImageNet32 [9]. In the default AG'S NEWS experiments, the pool has 50k training samples and 100k from Dbpedia [4]. Note that as with existing pool-based ME [34], **MExMI does not require the pool to contain training samples**. The main reason for such a mixed dataset composition is for us to evaluate the performance of MI [41] and show how much it can be enhanced by ME. In Section 5.4, we evaluate the performance of MExMI when the pool has no training sample at all.

## 5.2 Metric-based Shadow-Model MI

**Implementation Details**.  Recall that the hyper-parameters (such as epoch, initial learning rate, and optimizer) of shadow models $F_s$ can be adjusted to maximize the metric $Q$. Parameter $a$ in $Q$ is set as 0.05 and $f(\cdot)$ is set as $-log_{10}(\cdot)$.[3] The training dataset of $F_s$, denoted by $D^{(s)}$, is constructed by random sampling from $P$, and its size varies from the set $\{2000, 5000\}$. A non-training dataset, denoted by $D_n^{(s)}$, is also randomly sampled from $P \setminus D^{(s)}$. We implement two metric-based shadow-model MI methods in Section 4: *original shadow-model MI* [41] and *FSL shadow-model MI* that utilizes transfer learning [21] and data augmentation respectively. (See Appendix E.5 for experimental details.) For each MI method, we vary their hyper-parameters and thus $Q$ in various settings. An ideal shadow-model MI is built as a reference by using the same training dataset as the victim model.

**Results**.  We plot the MI attack models' accuracy with respect to shadow metric $Q$ in Fig. 2. We observe that $Q$ is positively correlated with the attack accuracy of shadow-model MI and therefore can guide the training process of the shadow models, whether using the original or FSL training algorithms. The recall rate of all MI attacks is almost $100\%$, so it is omitted from the figure. In addition, once the metric $Q$ is large enough ($\geq 6$), metric-based shadow-model MI attack models can achieve almost the same accuracy as the ideal shadow-model MI [39] even for 2k samples.

---

[3]In our experiments $l_s \in (0, 1)$ where $-log_{10}(\cdot)$ is non-negative and ever decreasing.

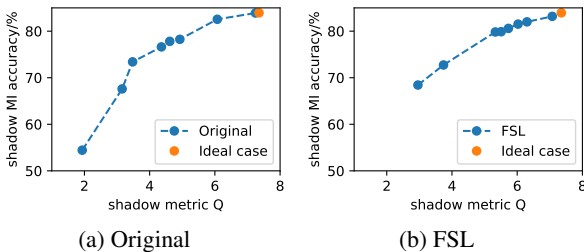

|             (a) Original             |             (b) FSL             |

Figure 2: Experiment results of metric-based shadow-model MI under different settings.

## 5.3 Overall Performance of MExMI

### 5.3.1 Implementation Details

We compare six PAME attacks, including four MExMI variants, namely, baseline ME without MI, Pre-Filter only, Post-Filter only, MExMI without semi-supervised boosting, regular MExMI (which adopts Mix-Match semi-supervised methods [5] for image classifiers, and consistency regularization [38] semi-supervised methods for text classifiers), and the ideal ME attack. The baseline ME is ActiveThief [34], which is the state-of-the-art PAME attack.[4] The ideal ME uses the real training samples as its pool. Moreover, to comprehensively evaluate MExMI, we added two additional state-of-the-art query-synthesizing-based ME baselines, including PRADA [26] and DFME [44] which share the same query budget with PAME attacks. Each MExMI variant has a MI result. In our experiments we compare our MI attacks that don't cost additional query budget with the existing MI attacks [39] that assume infinite query budget.

The AL algorithms used are entropy uncertainty [31], greedy k-center [40] and adversarial deep-fool [14] (see Appendix B for a brief explanation). The last algorithm is not evaluated on AG'S NEWS since there is no trivial way to adapt it to text classification. The copy models in MExMI share the same architecture as the victim models. The MI attack model $F_{MIA}$ is trained on initial seed samples. The preset weights ratio $\omega$ in MI Post-Filter is $5:1$. For CIFAR10 experiments, MExMI queries 2k samples in each round with a total of 8 rounds. For AG'S NEWS experiments, there are 6 rounds, each with 5k samples. All experimental results are the average measures of 5 trials. See Appendix E for more implementation details and Appendix F for supplementary experimental results.

### 5.3.2 Overall Results of MExMI

We use fidelity and model's accuracy obtained from the test datasets $D_t$ (CIFAR10 and AG'S NEWS test sets) to evaluate copy models against the victim model (see Section 2.2), and use accuracy, precision and recall of the inferred training datasets against ground truth to evaluate the MI accuracy.

Fig. 3 plots the fidelity of various PAME attacks with respect to iterations on CIFAR10 and AG'S NEWS (see more figures in Appendix F.1), respectively [5], and Table 2 shows the final results. Fig. 4 plots the accuracy, precision and recall of the MI attack of each MExMI variant. The results indicate that MExMI greatly boosts the potential of AL algorithms in PAME, and breaks the curse on query budget of existing MI. Overall, MExMI performs the best and achieves a fidelity gain of $7.76\%$, $7.8\%$, and $11.14\%$ on CIFAR10 over the baseline ME attack in all three AL methods. A similar gain of $4.46\%$ and $3.46\%$ is observed on AG'S NEWS over the baseline PAME attacks in both two AL methods. Without additional queries, the MI attacks of MExMI yield up to $83.20\%$ accuracy, $84.13\%$ precision and $75.93\%$ recall on CIFAR10, and $68.77\%$ accuracy, $71.73\%$ precision and $82.53\%$ recall on AG'S NEWS respectively, both on the par with existing MI attacks [39] that assume infinite query budgets.

**Impact of MI Post-Filter and MI Pre-Filter**.    In Fig. 3, on CIFAR10 attacks with MI Post-Filter always outperform those without it, by up to a $1.64\%$ increase on fidelity in the final results. The gain is more eminent in the beginning iterations, and then gradually decreases. On AG'S NEWS, MI

---

[4]INVERSENET [18] is another state-of-the-art work that adopts model inversion to assist ME. We do not include it in the experiments for two reasons. First, its performance is similar to ActiveThief under our 16k-query budget setting. Second, it augments query selection from the data pool with query synthesis from the model, which can be considered orthogonal to our method.

[5]The label 'MExMI w/o Boosting' in figures is an abbreviation for MExMI without semi-supervised boosting.

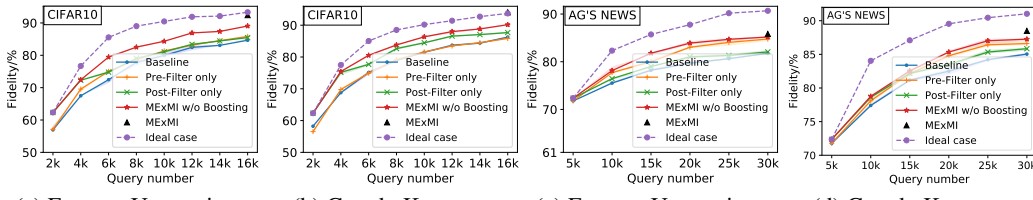

(a) Entropy Uncertainty    (b) Greedy K-center    (c) Entropy Uncertainty    (d) Greedy K-center

Figure 3: PAME results on CIFAR10 (16k budget) and AG'S NEWS (30k budget). Shadows represent error bars.

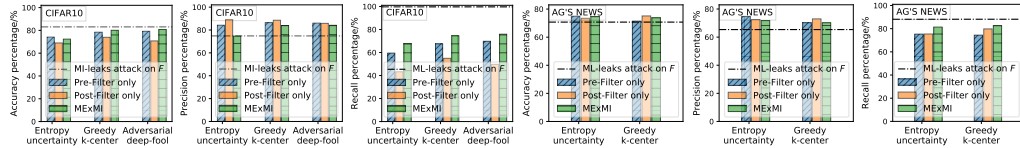

(a) MI Accuracy (b) MI Precision (c) MI Recall (d) MI Accuracy (e) MI Precision (f) MI Recall

Figure 4: MI attack results of MExMI. "ML-leaks" refers to the shadow-model MI attack in [39].

Post-Filter also performs effectively, with a maximum boost of $1.57\%$ on fidelity over those without it. As for MI Pre-Filter, except for the greedy k-center one in CIFAR10 experiments, attacks with Pre-Filter always outperform those without it. In addition, the gain does not decrease with more iterations, because the adversary data pool is much larger than the total query budget. Interestingly, we find that when MI Pre-filter and MI Post-filter work together, they can achieve a greater gain on fidelity than the sum of individual gains when they work separately. This suggests that **the two filters truly reinforce each other in our MExMI framework.**

Table 2: Results of Default PAME Experiments

| Fidelity (Accuracy)/% | CIFAR10 | | | AG'S NEWS | |
|---|---|---|---|---|---|
| | **Entropy Uncertainty** | **Greedy K-center** | **Adversarial Deep-fool** | **Entropy Uncertainty** | **Greedy K-center** |
| PRADA [26]/DFME [44] | 61.32 (60.12) / 11.20 (10.32) | | | - / 30.23 (25.00) | |
| Baseline(ActiveThief) | 84.65 (83.78) | 86.26 (85.69) | 82.93 (82.27) | 81.36 (78.66) | 85.03 (81.92) |
| Pre-Filter only | 85.38 (85.38) | 85.84 (85.22) | 86.17 (85.48) | 84.76 (81.57) | 86.61 (83.36) |
| Post-Filter only | 85.48 (84.71) | 87.68 (86.86) | 84.57 (84.00) | 82.06 (79.29) | 85.84 (82.38) |
| MExMI w/o Boosting | 89.10 (88.69) | 90.16 (89.21) | 90.14 (89.58) | 85.18 (81.98) | 87.26 (84.18) |
| MExMI | **92.41 (91.80)** | **94.06 (93.43)** | **94.07 (93.47)** | **85.82 (82.54)** | **88.49 (85.36)** |
| Ideal case | 93.31 (93.03) | 93.71 (93.38) | 93.66 (93.25) | 90.68 (87.51) | 91.03 (87.68) |

**Impact of Semi-Supervised Boosting.** In Fig. 3 and Table 2, MExMI outperforms MExMI without semi-supervised boosting by at least $3.11\%$ on CIFAR10 and $0.56\%$ on AG'S NEWS in terms of accuracy, which indicates that MExMI does benefit from effective MI attacks. A fidelity gain is observed in MExMI since the copy models' accuracy is closer to that of the victim model.

**MI Performance in MExMI.** The precision and recall of the adaptive shadow-model MI attacks of three MExMI variants are shown in Fig. 4. MExMI always performs the best and can achieve up to $83.20\%$ accuracy and $84.13\%$ precision on CIFAR10, and $68.77\%$ and $71.73\%$ precision on AG'S NEWS. This precision is even better than the state-of-the-art MI — ML-leaks [39] ($75.25\%$ on CIFAR10, $65.37\%$ on AG'S NEWS) which assumes unlimited query budget. Furthermore, our adaptive MI attacks have no additional cost when inferring training samples.

**Discussion About Potential Defenses.** There are two potential defenses against MExMI. First, MExMI is subject to MI-related defensive strategies that can reduce the accuracy of MI, such as using differential privacy [13], which in turn lower the fidelity of MExMI. Second, as with other model extraction attacks, MExMI can also be defended by model provenance techniques, such as watermark embedding [24], to detected copyright infringement from an extracted model.

## 5.4 Impact of Adversary Data Pool on PAME

In this experiment, we study how the quality of adversary data pool affects the outcome of PAME attacks. Since there are cases where the data pool does not contain any training data, the results of MI attack are not evaluated. To be fair, we fix the size of the pool to 150k samples and change the proportion of training data in it to vary its quality. Since the total training samples are 50k, this proportion is capped at $1/3$.

The results on CIFAR10 are shown in Table 3. We can see that the quality of the data pool greatly affects each PAME attack. MExMI consistently outperforms the baseline irrespective of the quality, even in the complete absence of victim's training data, i.e., when the adversary has no access to the true training samples. In such extreme case, we also observe that Pre-Filter only is outperformed by the baseline. This is due to the fact that the Pre-Filter cannot find any training data in the pool and thus excludes most of them for training. Since the filtered data pool is too small, active learning might not be effective.

Table 3: Impact of the Adversary Data Pool on PAME Attacks with 16k Query Budget

| The Proportion of $P_v$ in $P$ | | Baseline | Pre-Filter only | Post-Filter only | MExMI w/o Boosting | MExMI |
|---|---|---|---|---|---|---|
| **Fidelity (Accuracy)/%** | **0%** | 76.30(75.62) | 74.23(73.93) | 76.29(75.71) | 77.40(76.80) | **79.11(78.51)** |
| | **25%** | 81.07(80.48) | 83.91(83.50) | 83.55(83.05) | 89.99(88.78) | **92.91(91.96)** |
| | **33.33%** | 82.93(82.27) | 86.17(85.48) | 84.57(84.00) | 90.14(89.58) | **94.07(93.47)** |

## 5.5 Impact of Output Access

We investigate the impact of output access granted to our PAME attacks. We limit the output access to top-1 score and show the results on CIFAR10 in Table 4. It indicates that even with limited output access, the three modules of MExMI still perform consistently well. Among various PAME attacks, MExMI always performs the best and can outperform the baseline by $11.47\%$ on fidelity.

Table 4: Impact of Output Access on PAME Attacks

| **Fidelity (Accuracy)/%** | | Baseline | Pre-Filter only | Post-Filter only | MExMI w/o Boosting | MExMI |
|---|---|---|---|---|---|---|
| **Output Access** | **Probabilities** | 82.93(82.27) | 86.17(85.48) | 84.57(84.00) | 90.14(89.58) | **94.07(93.47)** |
| | **Top-1 Scores** | 79.53(79.00) | 81.89(81.33) | 81.94(81.18) | 85.60(84.89) | **91.00(90.40)** |

## 5.6 Case Study: Blackbox Attacks on MLaaS ModelArts

We use a real case study to show the feasibility and real-life impact of MExMI. We target at *ModelArts*, the MLaaS provided by Huawei Cloud [3] where developers can train and deploy their ML models in the cloud, and then access them via Web API (e.g., CURL). We train and deploy a classification model on ModelArts without the knowledge of its architecture using SVHN and then perform various PAME attacks on it with VGG16. See Appendix E.6 for more implementation details.

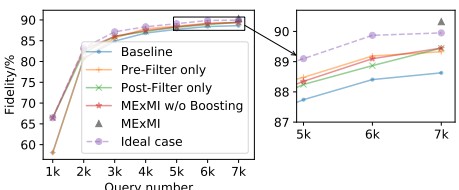

Figure 5: PAME attacks' results in ModelArts on SVHN (7k budget).

The experimental results of the adversarial deep-fool PAME are shown in Fig. 5. All four MExMI variants outperform the baseline. MExMI achieves the highest $90.32\%$ fidelity and MExMI without semi-supervised boosting comes the second with $89.45\%$ fidelity. The final MI results of MExMI framework are shown in Table 5. We observe similar precision and recall of the three variants, all on the par with ML leaks [39],

Table 5: MI attacks' results in ModelArts

| MI Attacks | Precision/% | Recall/% |
|---|---|---|
| Pre-Filter only | 86.94 | 91.69 |
| Post-Filter only | 87.17 | 90.50 |
| MExMI | 87.78 | 91.77 |
| ML-leaks on $F$ | 87.83 | 92.38 |

the state-of-the-art unsupervised MI attack that exhausts all pool data, which costs ten times higher.

# 6   Related Works

**Model Extraction**.   Model extraction (ME) on MLaaS models has drawn a lot of attention since Tramèr, et al. (2016) [43], mainly because it can be used for downstream attacks. In terms of query data availability, ME can be categorized into query-synthesizing ME and query-acquiring ME. The former is used when the adversary does not possess enough real data for query, which includes iterative active frameworks [35, 26] and minimax-game frameworks [44, 27]. Despite saving the data collection cost, it suffers from huge query budget demands. The latter is used when the adversary spares extra cost to actively collect real data for query [10, 23, 34, 10]. We summarize the above ME attacks in Table 6, and highlight the data pool requirement of each work and their dependency on data selection and data synthesis.

**Memership Inference**.   The vulnerability of MLaaS models' training data privacy has raised great conerns [51, 22] and various attacks against it are proposed. Of these, Membership inference (MI) which attempts to infer the training membership of some query data, shows a worrying breach of data privacy. Existing works exploit different sources of information such as outputs [41, 39] and model parameters [30] in either black-box or white-box setting to infer the training data. In black-box attack scenario, moet MI works follow shadow-model MI [41, 39] which cast the attack to a supervised learning problem. Another type of black-box MI is to utilize the unsupervised learning [39, 37] with different measure metrics, such as output entropy, predicted confidence [39] and loss [37]. On the other hand, a white-box MI is proposed [30] which infers the training data and non-training data distribution characteristics via the victim model's parameters. More recent works focus on extending MI attacks to less favorable scenarios, for example when the model output is label only [8, 30], and extremely low query budget.

Table 6: Existing Learning-based Model Extraction Works

| Attacks | Data Pool Requirements | Data Synthesis | Data Selection |
|---|---|:---:|:---:|
| Tramèr (2016) [43] | None | ✓ | |
| Knockoff (2019) [33] | Probelm Domain, or Public Pool | | ✓ |
| Papernot (2017) [35] | Probelm Domain Initial set | ✓ | |
| PRADA (2019) [26] | Probelm Domain Initial set | ✓ | |
| DFME (2021) [44] | None | ✓ | |
| ActiveThief (2020) [34] | Public Pool | ✓ | |
| Jagielski (2020) [23] | Probelm Domain Pool | | ✓ |
| INVERSENET (2020) [18] | Public Pool | ✓ | ✓ |
| MExMI (this work) | Public Pool | | ✓ |

# 7   Conclusion

In this paper, we proposed a PAME crossover MI framework called MExMI where the model and training data privacy can trigger a chain reaction to boost the performance of both attacks. The framework is generic in that it can adopt various PAME and MI attacks. In our experiments, MExMI improves the fidelity of copy models to $94.07\%$, up from the basic PAME by $11.14\%$. Meanwhile, the MI accuracy and precision can reach $83.20\%$ and $84.13\%$ without additional query budget, on par with state-of-the-art MI attack which requires about 10 times more queries.

## Acknowledgments and Disclosure of Funding

This work was supported by National Natural Science Foundation of China (Grant No: 62072390, 62102334), the Research Grants Council, Hong Kong SAR, China (Grant No: 15222118, 15218919, 15203120, 15226221, 15225921, and C2004-21GF), and a Huawei research grant (TC20200831001).

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
