# Appendix

## A MExMI Algorithm

---

**Algorithm 1** MExMI attack

---

1: **Input:** $F$, $P$;  **Parameters:** $k$, $b$, $\omega$;
2: **Output:** $F'$, $\hat{D}$;
3: $[\mathbf{x}_1, \ldots, \mathbf{x}_k]_0 \leftarrow \mathtt{RandomPick}(P)$;
4: $P_r \leftarrow P \setminus [\mathbf{x}_1, \ldots, \mathbf{x}_k]_0$, $d \leftarrow d - k$, $i = 0$;
5: $D_q \leftarrow [\{\mathbf{x}_1, F(\mathbf{x}_1)\}, \ldots]_0$;
6: $F_{MIA} \leftarrow \mathtt{MIUpdate}(D_q, P)$;       ▷ Section 4
7: **while** $b >= 0$ **do**
8:   $D_{q+}, D_{q-} \leftarrow F_{MIA}(D_q)$;
9:   $\mathtt{AssignWeigtht}(D_{q+}, D_{q-}, \omega)$;
10:   $F'_i \leftarrow \mathtt{WeightLossTrain}(D_{q+}, D_{q-})$;
11:   $P_t \leftarrow F_{MIA}(F'_i, P_r)$;
12:   $[\mathbf{x}_1, \ldots, \mathbf{x}_k]_{i+1} \leftarrow \mathtt{ActiveLearning}(P_t, F'_i)$;
13:   $P_r \leftarrow P_r \setminus [\mathbf{x}_1, \ldots, \mathbf{x}_k]_{i+1}$,
14:   $D_q \leftarrow [\{\mathbf{x}_1, F(\mathbf{x}_1)\}, \ldots]_{i+1} \cup D_q$, $d \leftarrow d - k$, $i \leftarrow i + 1$;
15: **end while**
16: $\hat{D} \leftarrow F_{MIA}(F', P)$;
17: $F' \leftarrow \mathtt{SemiSupervisedTrain}(D_q, \hat{D})$;

---

## B Pool-based Active Learning Algorithms

MExMI can use various pool-based AL algorithms available in the literature. In this paper, we focus on three AL algorithms based on different metrics of samples: uncertainty, diversity, and vulnerability to deep-fool perturbation. As AL algorithms often calculate the distance between samples [50], MExMI also provides an encoding process to reduce dimension for high-dimensional feature space. In essence, it uses $F'_{i-1}$, the copy model trained in the previous iteration, as an encoder. When inter-sample distances are required, we use inter-vector distances between the outputs of the encoder for calculation. The distance is calculated using the output probability vectors of $F'$ whenever the calculation of the distance is needed.

### B.1 Entropy Uncertainty

One of the most common ways to measure uncertainty is entropy [31]. The larger the entropy, the higher the uncertainty level is. For a sample $\{\mathbf{x}, \hat{\mathbf{y}}\}$ in the data pool, where $\hat{\mathbf{y}} = F'(\mathbf{x}) = [\Pr(1|\mathbf{x}), \Pr(2|\mathbf{x}), \ldots, \Pr(K|\mathbf{x})]^T$, its entropy is defined as

$$\Phi_{ent}(\mathbf{x}) = -\sum_{k=1}^{K} \Pr(k|\mathbf{x}) log(\Pr(k|\mathbf{x})), \tag{1}$$

where $K$ is the number of class labels.

### B.2 Greedy K-center

One classic diversity-based AL is the greedy k-center algorithm [40]. Let $D_q := [\{\mathbf{x}_q, F(\mathbf{x}_q)\}, \ldots]$ denote the set of samples selected previously, and $P := [\mathbf{x}_p, \ldots]$ the data pool. Greedy k-center algorithm sets $[F'(\mathbf{x}_q), \ldots]$ as cluster centers and selects the sample $\mathbf{x}_s$ that has the largest Euclidean distance from all existing centers. Formally,

$$\mathbf{x}_s = argmax_{\mathbf{x}_p \in P}\{min_{(\mathbf{x}_q, \mathbf{y}_q) \in D_q} \|F'(\mathbf{x}_p) - F'(\mathbf{x}_q)\|_2^2\}. \tag{2}$$

Then we query the selected sample $\mathbf{x}_s$ and update $D_q$ by $D_q = D_q \cup \{\mathbf{x}_s, \mathbf{y}_s\}$, where $\mathbf{y}_s = F(\mathbf{x}_s)$.

## B.3  Adversarial Deep-Fool

Deep-Fool based AL (DFAL) uses sample perturbation attack in Deep Fool to calculate the distance between samples and decision boundaries, a.k.a., margin, and then selects those with the smallest perturbation distance to query [14]. The perturbation algorithm iteratively perturbs samples by adding linear noise until the samples are misclassified by the copy model $F'$.

## C  Proof of Theorem 1

**Proof C.1** *For a multi-classifier with $K$ labels and $n$ training samples, the training loss $l$ is measured using cross-entropy:*

$$l = -\frac{1}{n} \sum_{\boldsymbol{x} \in \mathbb{X}} \sum_{y=1}^{K} \Pr\left(Y = y | \boldsymbol{x}\right) log(\Pr\left(\hat{Y} = y | \boldsymbol{x}\right)), \tag{3}$$

*where $Y$ is the ground truth label variable, and $\hat{Y}$ is the predicted label variable. Since the ground truth probability vectors are in one-hot format, the training loss in the $j$-th class, $l_j$, can be rewritten as:*

$$l_j = -\frac{1}{n} \sum_{\boldsymbol{x} \in \mathbb{X}_j} log(\Pr\left(\hat{Y} = j | \boldsymbol{x}\right)) \tag{4}$$

*Therefore, the gap of loss in the $j$-th class between the distribution of the shadow model $\mathbb{Y}^{(s)}$ and that of the victim model $\mathbb{Y}$ is:*

$$\Delta l_j = \frac{1}{n^{(s)}} \sum_{\boldsymbol{y}^{(s)} \in \mathbb{Y}_j^{(s)}} log(y_j^{(s)}) - \frac{1}{n} \sum_{\boldsymbol{y} \in \mathbb{Y}_j} log(y_j), \tag{5}$$

*where $y_j$ represents the $j$-th element of $\boldsymbol{y}$. On the other hand, We define $b'_{jj}$ as the log-bias between the two logarithm distributions for class $j$:*

$$b'_{jj} \equiv \frac{1}{n_j^{(s)}} \sum_{\boldsymbol{y}^{(s)} \in \mathbb{Y}_j^{(s)}} log(y_j^{(s)}) - \frac{1}{n_j} \sum_{\boldsymbol{y} \in \mathbb{Y}_j} log(y_j). \tag{6}$$

*From Eqn. 7 and 8, we get:*

$$b'_{jj} = \frac{\Delta l_j}{a_j}, \tag{7}$$

*where $a_j$ is the proportion of class $j$ in both shadow and victim model training sets. $\boldsymbol{b}_j \in \mathbb{R}^K$ is a vector, and its element in dimension $j$ is denoted by $b_{jj}$.*

*To complete the proof, in Eqn.9 we need to replace $b'_{jj}$ with $b_{jj}$, the $j$-th element in vector $\boldsymbol{b}_j$. This replacement is correct because of the following two assumptions, whose validity will be verified experimentally in Section 5.*

**Assumption 1**. *The correlation coefficient $\rho$ between a distribution and its logarithm distribution is positive. Formally,*

$$\rho_{|b'_{jj}|, |b_{jj}|} > 0, \tag{8}$$

*where $j \in [1, \ldots, K]$.*

**Assumption 2**. *For two models with the same structure and hyper-parameters, for any class $j$, the correlation coefficient $\rho$ between the output vector distribution and its $j$-th element is positive. Formally,*

$$\rho_{|b_{jj}|, |\boldsymbol{b}_j|} > 0, \tag{9}$$

*where $j \in [1, \ldots, K]$.*

*Based on the above two assumptions, we replace $b'_{jj}$ with $b_{jj}$ in Eqn.9 and obtain $\Delta l_j \propto \boldsymbol{b}_j$.*

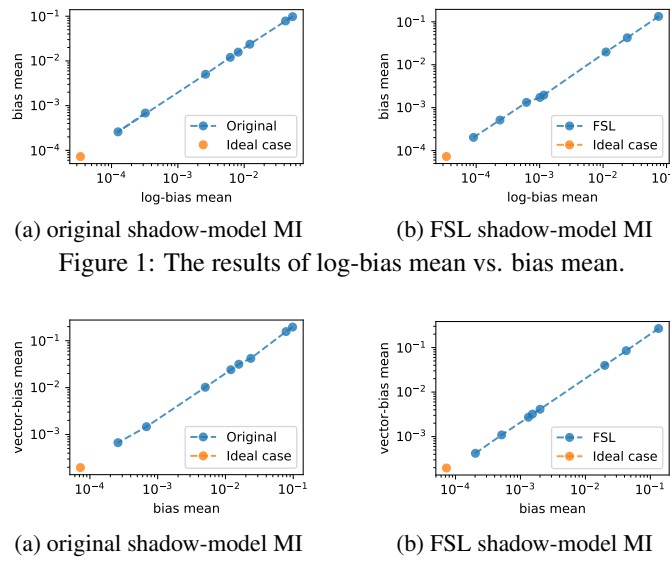

(a) original shadow-model MI      (b) FSL shadow-model MI

Figure 1: The results of log-bias mean vs. bias mean.

(a) original shadow-model MI      (b) FSL shadow-model MI

Figure 2: The results of bias mean vs. vector-bias mean.

To verify Assumptions 1 and 2 of Theorem 1, we calculated the *bias mean* as the mean of bias of all classes, i.e., $\sum_{j=1}^{K} |b_{jj}|$, the *log-bias mean* as the mean of log-bias of all classes, i.e., $\sum_{j=1}^{K} |b'_{jj}|$, and the *vector-bias mean* as the mean of scalar bias $\sum_{dim} \sum_{j=1}^{K} |\mathbf{b}_j|$ of all classes. The results are shown in Fig. 6 and Fig. 7 respectively. We observe that:

1. The bias mean is positively correlated with the log-bias mean, which justifies Assumption 1.

2. The vector-bias mean is positively correlated with the bias mean, which justifies Assumption 2.

## D   Few-shot Learning (FSL) Shadow-model MI

In MExMI, we use the following two FSL approaches to train shadow models, leading to two metric-based shadow-model MI.

**Data-augmented shadow-model MI**. Data augmentation expands the training dataset to accelerate convergence by adding synthetic samples transformed from existing samples. In image classification, an image can be flipped, panned or rotated to enrich the training set [29].

**Transfer shadow-model MI**. Transfer learning shares the knowledge from a source domain to reduce the training complexity in a target domain. In image classification, NN models are suitable for transfer learning, because their shallow layers learn task-independent abstract features, and deep layers are more task-related [21]. As such, an adversary can transfer the shallow layers of a pre-trained NN model to initialize a shadow model's parameters and to accelerate its convergence. This technique is valid even if the pre-trained model's problem domain is different from that of the shadow models.

## E   Implementation of Experiments

### E.1   Datasets

**CIFAR10**. CIFAR10 is an image dataset in color (with 3 channels) with 10 class labels, 50k training samples, and 10k test samples. The image samples have a resolution of 32 and are evenly distributed into 10 classes. It is a well-known benchmark dataset to evaluate image classifiers.

**SVHN**. SVHN is another 32-resolution benchmark for color image classification. It consists of street-view images of door numbers, which are labelled with digits from "0" to "9". The dataset contains 73,257 images for training, and 26,032 images for testing.

**AG'S NEWS**. AG'S NEWS is a benchmark for text classification. It consists of titles and descriptions of articles from 4 news classes, namely "World", "Sports", "Business" and "Sci/Tech". This dataset contains 120k training samples and 7.6k test samples.

## E.2 Running Environment

Experiments are implemented with Python 3.7 on a desktop computer running Windows 10 with AMD Ryzen 7 2700X CPU and 64 GB RAM.

## E.3 Implementation Details for ML-leaks [39] Membership Inference Attacks

In the overall evaluation experiments and case study experiments, our membership inference (MI) attacks on the copy models are compared with ML-leaks [39] MI direct attacks on the victim models. For the shadow-model MI attack in ML-leaks, we use one shadow model trained on the victim model's training data set to build its MI attack model. For unsupervised MI attack in ML-leaks, 1000 non-member noise samples are queried from the victim model. The parameter tolerant percentage $t$ is set to 0.06 in the threshold decision method.

## E.4 Implementation Details for Iterative MExMI attacks.

The hyper-parameters are set as follows: *initial learning rate* (*lr*)=0.03 for image classifiers, *lr*=0.01 for text classifiers, *momentum*=0.5, *weight decay*=$10e-4$, *epochs*=150. The optimization method is SGD accelerated by Nesterov Gradient Method [12]. The *lr* scheduler multiplies *lr* with the factor 0.1 at epoch 100. To be fair, all attacks, including the ideal one, use the same initial seed samples. The adaptive shadow-model MI used in MI Pre-Filter and MI Post-Filter is trained on initial seed samples for 150 epochs.

## E.5 Details of Metric-based Shadow-model MI Experiments

For transfer shadow-model MI, it needs prior knowledge of the source model. In the experiment, we use a 5-block model with the same shallow structure as the victim model as the source model and train it on CIFAR100. We then transfer the parameters of the three shallow blocks to initialize the shadow models. For data-augmented shadow-model MI, we add augmented samples to the training dataset to double its size. We adopt the same augmentation policy as in [49], which includes inverting, rotating, sharpness etc., but excludes those methods used to train models for a fair comparison. The test dataset consists of 10k victim's training images as positive samples and another 10k non-training images as negative samples.

## E.6 Implementation Detail for Case Study

We train and deploy a classification model on ModelArts without the knowledge of its architecture using SVHN and then perform various PAME attacks on it. Our adversarial data pool consists of 2.5k SVHN and 5k ImageNet32 images as we have to measure both ME results and MI results of MExMI. Since ModelArts returns top-5 probabilities with three decimal places, we choose unsupervised MI as the MI module in MExMI. The parameter tolerant percentage $t$ is set to 0.06 in the threshold decision method [39]. The architecture of the copy model is VGG16 with batch normalization. The query budget is 7k, and the size of initial seeds as well as the step are both 1k.

# F  Supplementary Experiments

## F.1 The Overall Performance MExMI on Adversarial Deep-fool

We present here a supplement to the performance of MExMI in Adversarial Deep-fool's PAME during the iterative process, as shown in Fig. It is a supplement to Fig. 3.

## F.2 The Evaluation of MI Pre-Filter in the Iteration

To understand the underlying mechanism why MI Pre-Filter works, we track the filtering results of Pre-Filter in each iteration of MExMI attack on CIFAR10. The results are shown in Fig. 9. We

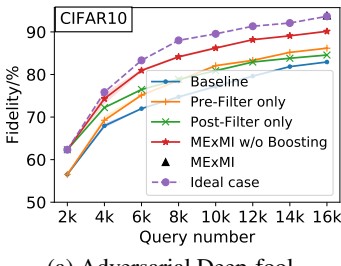

(a) Adversarial Deep-fool

Figure 3: PAME attacks' results on CIFAR10 during the iteration.

observe that MI Pre-Filter can accurately find victim's training samples in the remaining pool, so the training set for copy model is gradually restored through iterations.

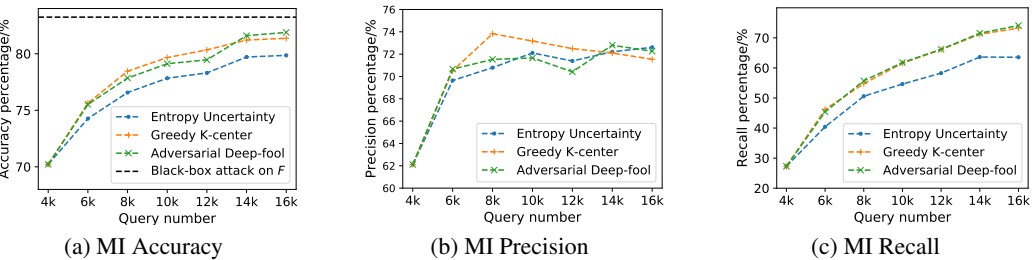

(a) MI Accuracy (b) MI Precision (c) MI Recall

Figure 4: Performance of MI Pre-Filter in MExMI on CIFAR10.

### F.3 Transferability of Adversarial Attacks

We measured the transferability of adversarial samples obtained from the FSGM [19] adversarial attacks (at a rate of $\epsilon = 0.1$) on the PAME copy models. The transferability rate is the fraction of these samples misclassified by the victim model. The results on CIFAR10 are shown in Table 7. MExMI consistently has higher transferability rate than the baseline, indicating that our MExMI algorithms is also superior from this perspective.

Table 1: Transferability of FGSM attacks

| Active Learning | Transferability / % | | |
| --- | --- | --- | --- |
| | **Entropy Uncertainty** | **Greedy K-center** | **Adversarial Deep-fool** |
| Baseline | 51.76 | 57.47 | 57.59 |
| Pre-Filter only | 55.96 | 59.72 | **60.99** |
| Post-Filter only | 59.17 | 62.53 | 57.90 |
| MExMI | **63.87** | **62.66** | 58.57 |

### F.4 Impact of Weight Ratio in MI Post-Filter

In MI Post-Filter, we introduce a weight ratio $\omega$ between loss weights of the inferred training data and non-training data, which has an effect on both Post-Filter only and MExMI without semi-supervised boosting variants. In this experiment, we vary $\omega$ in CIFAR10 experiments and show the results in Table. 8. We observe that $\omega$ has a very limited impact on the fidelity and therefore our MExMI framework is robust to this parameter.

Table 2: Impact of Post-Filter Weight Ratio

| Weight Ratio | Fidelity (Accuracy) / % | | |
| --- | --- | --- | --- |
| | **3:1** | **5:1** | **7:1** |
| Post-Filter only | 85.12 (84.20) | 84.57 (84.00) | 84.59 (83.93) |
| MExMI w/o Boosting | 89.61 (99.84) | 90.14 (89.58) | 90.21 (89.61) |

Table 3: Performance Boosting Using Different ML Optimization Methods

| Methods | Fidelity (Accuracy) / % | | |
|---|---|---|---|
| | Baseline | Baseline + Data-Aug | Baseline + Data-Aug + Ensemble |
| Entropy Uncertainty | 89.10 (88.69) | 90.10 89.97) | **91.57 91.36)** |
| Greedy K-center | 90.16 (89.21) | 91.11 (90.98) | **92.86 (92.60)** |
| Adversarial Deep-fool | 90.14 (89.58) | 91.32 (90.80) | **92.84 (92.32)** |

## F.5  The Ability of Evading PRADA Defence

For image classification, PRADA[26] is the state-of-the-art method to detect ME attacks. The detection is based on the distribution of consecutive query data, as PRADA believes that the adversary tends to issue query samples across an exceptional feature space. To evaluate the ability of MExMI evading such detection, we measure the minimal L2 distance between query samples of MExMI and several benchmark ME attacks on CIFAR10. The results are shown in Fig. 10. We can see that MExMI and the baseline attack are not different from benign queries (subjecting to Gaussian distribution) and both cannot be detected by PRADA. In contrast, the distributions of PRADA attack[26] and Black-box DNNs[35] have distinct traits, which are thus easier to be detected.

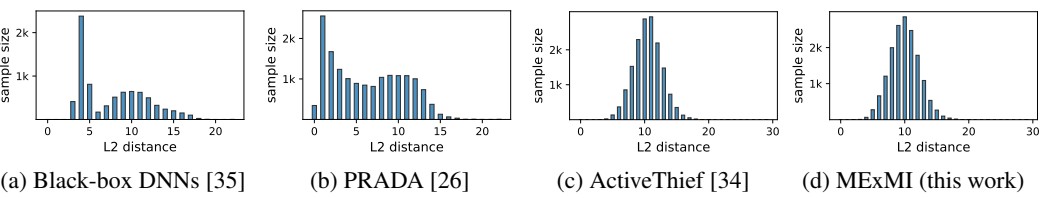

(a) Black-box DNNs [35]     (b) PRADA [26]     (c) ActiveThief [34]     (d) MExMI (this work)

Figure 5: Distribution of L2 distance required in PRADA defence.

## F.6  Impact of ML Optimizations

As there are many emerging optimization methods in ML, in this subsection we investigate what impact they have on PAME attacks. In particular, we focus on the following two methods:

- **Data augmentation.** It is used in the training process to prevent overfitting. This method has become increasingly popular in the domain of image classification [36]. As shown in the experiments below, applying data augmentation in PAME can significantly improve the fidelity.
- **Ensembles for neural networks.** Ensembling a set of models trained separately is well known for effectively reducing generalization error [20]. As shown in the experiments below, applying ensembles in PAME can improve the fidelity.

We use the performance results in Section 5.3 on CIFAR10, especially MExMI without semi-supervised boosting, as the baseline in this experiment. We then use the transforming policy in [49] to perform a richer data augmentation and model averaging ensemble method. The results are shown in Table 9, where "Data-Aug" denotes data augmentation. Richer data augmentation improves baselines' fidelity by $1.18\%$, and the ensemble method further improves fidelity by another $1.52\%$ to $92.84\%$. These results warn us that in a never-ending battle between model owners and model extractors, emerging technologies in ML may favour the latter rather than the former.