# OpenReview forum: "MExMI: Pool-based Active Model Extraction Crossover Membership Inference"
_NeurIPS.cc/2022/Conference — NeurIPS 2022 Accept_

### Official Review · Reviewer_eaMb · 2022-06-22

**Rating:** 5
**Confidence:** 4
**Soundness:** 3 good
**Presentation:** 3 good
**Contribution:** 3 good

**Summary:**

In this paper, the authors propose a method to boost model extraction attacks. Specifically, they propose to use MIA (membership inference attack) to identify homogeneous samples in the adversary data pool and make full use of them to improve the effectiveness of model extraction.

**Questions:**

Please refer to Cons.

**Limitations:**

Please refer to Cons.

**Strengths And Weaknesses:**

Pros:

1. The authors proposed a method to boost model extraction attacks under the scenario that the attacker has no knowledge of the victim training dataset.

2. The idea of using membership attacks to boost model extraction attacks is novel.

Cons:

1. The author mentioned that the proposed attack does not require the pool to contain training samples. However, the key to the proposed attack is to use MIA (membership inference attack) to infer the victim's training data (or the distribution of the training data) and use these data to perform a model extraction attack. If the pool does not contain the victim training data, the MIA success rate will be low, and the effect of boosting model extraction boost is limited.

2. The attack setting about the adversary pool is impractical. In practice, the adversary always cannot guarantee that the adversary data pool contains the victim training data (especially containing all the victim training data). The assumption that the victim training data accounts for 1/3 of the adversary pool is not reasonable. The attack process is more like the adversary first finds the victim training data through MIA and trains the surrogate model with it.

3 The author only considered ActiveThief as the baseline. Although the proposed method is aimed at boosting the model stealing attack based on active learning, it should also be compared with other types of model stealing attacks (such as PRADA [1], KnockoffNet [2], etc.) in query efficiency.

[1] Mika Juuti, Sebastian Szyller, Samuel Marchal, and N Asokan. Prada: protecting against dnn model stealing attacks. In Proceedings of Euro S&P, pages 512–527, 2019.
[2] Tribhuvanesh Orekondy, Bernt Schiele, and Mario Fritz. Knockoffnets: Stealing functionality of black-box models. In Proceedings of CVPR, pages 4954–4963, 2019.

---

> ### Author Response · Authors · 2022-08-02
> **Author Response to the Questions of Reviewer eaMb**
>
> **1, About the training sample assumption in the adversary pool and the role of MI in MExMI**
>
> Although MI is originally designed to infer the training samples, in MExMI it goes beyond “exact” training samples and covers “homogeneous” samples as well. The latter serves similar purpose to assist model extraction (ME). In fact, the experimental results in Section 5.4, Table 3 shows that MExMI has a good fidelity gain over the baseline ME attack (79.11% vs. 76.30%) even when there is no training sample in the pool, which shows the effectiveness of homogeneous samples found by MI. As another example, semi-supervised boosting in MExMI achieves 79.11% fidelity, as opposed to 62.19% if MI is not applied in the pool without any training sample.
>
>
> **2, The victim training data accounts for 1/3 of the adversary pool in the overall experiments.**
>
> It is a common practice in model extraction attacks, such as PRADA [23], Papernot [43], and Jagielski [21], to use a subset or even the entire training dataset as the initial pool. We believe this assumption/requirement is more stringent than ours as MExMI does not need to know exactly which ones are training samples and which ones are non-training samples. Furthermore, as responded above, in Section 5.4, our pool does not contain any training samples and MExMI still outperforms baseline ME attacks without MI filters.
>
> **3, New model extraction baselines**
>
> In the revision, we have added more state-of-the-art model extraction attacks as baselines in Section 5.3.2 Table 2, including PRADA [23] and DFME [44]. Knockoff [32] is excluded due to its strong assumptions about the pool’s coarse-to-fine hierarchy (e.g., the top-down nodes of a tree is animal, bird, and sparrow) for reinforcement learning. In terms of query efficiency, the new results show that MExMI has a significant advantage over all baselines. For example, in the image classification experiments, with a 16K query budget, the fidelity of DFME is 11.20%, PRADA is 61.32%, and MExMI is 94.07%.

---

> > ### Comment · Reviewer_eaMb · 2022-08-05
> > **Response to rebuttal**
> >
> > Thank you for clarifying some of the questions. Most of my concerns have been addressed and I would like to update my rating to 5.

---

> > > ### Author Response · Authors · 2022-08-05
> > > **Thank You for Updating Your Review**
> > >
> > > We are pleased to hear that we were able to address most of your concerns and received an updated rating. If there are still concerns / open questions, we would be happy to hear and discuss them.

---

### Official Review · Reviewer_LJzA · 2022-07-11

**Rating:** 5
**Confidence:** 4
**Soundness:** 2 fair
**Presentation:** 3 good
**Contribution:** 3 good

**Summary:**

This paper proposes a new pool-based active model extraction (PAME) attack. The key observation is that in PAME, the data pool that the attacker has access to may be different from the data samples used to train the model, which can hurt attack performance. Based on this observation, the paper proposes an iterative attack framework where a membership inference (MI) attack is utilized to identify training samples in the data pool, which is used to improve the extracted model through active learning. In turn, the extracted model can be used to improve the accuracy of membership inference. It is shown through experiments that MExMI outperforms ActiveThief [33], a PAME attack, in terms of model fidelity and obtains similar MI performance as an existing MI attack [38] that has an unlimited query budget.

**Questions:**

What is the ideal ME attack considered in the experiments? According to Table 2, MExMI can sometimes outperforms the ideal ME attack. How is that possible?

Table 3 shows that even when the data pool does not contain any training samples, MExMI still outperforms ActiveThief, although the improvement is rather limited. Is the improvement because of MI or something else?



**Limitations:**

It would help to briefly discuss potential defenses against the proposed attack.

**Strengths And Weaknesses:**

Both ME and MI have been intensively studied recently. The observation that the attacker may not have access to the victim's training data in ME attacks is not new. But the idea of letting ME and MI reinforce each other is new. To achieve this, the paper has adapted shadow-model MI attacks to the setting when there is a query budget and when queried samples are not drawn from the same distribution as the victim's training samples. Although these adaptations are mainly straightforward, they are clearly explained and seem effective according to the experiment results. That being said, I have two concerns about the paper.

First, the paper considers a relatively simple setting when no defense is applied for either ME or MI. Given that various defenses have been proposed for both attacks, it is important to understand if ME and MI can still reinforce each other effectively in the presence of defenses. In particular,  it is unclear if the adaption proposed for shadow-model MI still works in the presence of defenses against MI. Note that without defenses, ActiveThief already obtains a very good ME attack performance. Although the experiments show that MExMI can do even better, the results are not very impressive from a practice point of view. It would be much more interesting if similar results could be shown when state-of-the-art defenses are applied.

Second, I also have some doubts about the exact role that MI plays in MExMI. According to the introduction and the example in Table 1, the MI component is used to infer training samples from the data pool and best utilize them. However, it is claimed in the experiments that MExMI does not require the data pool to contain training samples. Table 3 shows that even when the data pool does not contain any training samples, MExMI still outperforms ActiveThief, although the improvement is rather limited. Is the improvement because of MI or something else? Note that the MExMI attack becomes less useful from a practice point of view if the MI component only helps when the data pool contains a certain amount of training data, as the attacker typically has no access to true training samples in ME attacks. The proposed attack would be more useful if the MI component could help identify data samples that can best assist ME even if they are not exactly the same as training samples.

---

> ### Author Response · Authors · 2022-08-02
> **Author Response to the Questions of Reviewer LJzA**
>
> **1, How does MExMI behave in the presence of defenses against MI?**
>
> We have conducted a preliminary experiment on adversary deep-fool active learning (i.e., ActiveThief). As any MI defense method eventually aims to reduce the accuracy of MI attacks, in the experiment we replace the original MI module (83.08% accuracy) with a low-accuracy MI module (73.83% accuracy) to emulate the presence of MI defenses. The fidelity of MExMI slightly drops from 94.07% to 91.62% for a 10% drop of MI accuracy, and still significantly outperform the original ActiveThief (82.93% accuracy). This shows that MExMI can still work very well in the presence of defenses against MI.
>
> **2, The exact role of the MI module in MExMI and its descriptions.**
>
> Thanks for your constructive suggestions. To more precisely describe the role of the MI module in MExMI, we have changed “utilizing MI to find the exact training data” to “utilizing MI to find homogenized samples of those training data”, to assist model extraction (ME).
>
> We have revised Table 3 to adopt semi-supervised learning and thus rule out the possible gain of MExMI due to the training algorithms. When the pool contains no training samples at all, MExMI still achieves 79.11% fidelity whereas ActiveThief with semi-supervised learning can only achieve 62.19%. The results show that MExMI makes good use of the homogeneous data found by the MI module.
>
> **3, What is the ideal ME in the experiments?**
>
> The “ideal ME” uses the victim’s training samples as the adversary pool (Section 5.3.1). In other words, the ideal ME is a “whitebox” ME attack when the victim’s training samples are completely known.
>
> The results of MExMI can sometimes be even higher than the ideal ME because the latter does not have a semi-supervised training process.
>
> **4, limitations: discuss potential defenses against the proposed attack.**
>
> There are two potential defenses against the proposed attack. First, MExMI is subject to MI-related defensive strategies that can reduce the accuracy of MI, which in turn lower the fidelity of MExMI. Second, as with other model extraction attacks, MExMI can also be defended by model provenance techniques, such as watermark embedding [J21], to detected copyright infringement from an extracted model.
>
> _[J21] Jia, Hengrui, et al. "Entangled watermarks as a defense against model extraction." 30th USENIX Security Symposium (USENIX Security 21). 2021._

---

> > ### Comment · Reviewer_LJzA · 2022-08-07
> > **Response to rebuttal**
> >
> > Thanks for clarifying some of my concerns. It's unclear to me how the low-accuracy MI module was emulated. How did you pick the accuracy (73.83%) and how was it realized on the dataset? Why is this considered to be representative of the actual behavior of MI under defense?

---

> > > ### Author Response · Authors · 2022-08-08
> > > **Thank You and Response to The New Comments**
> > >
> > > Thanks for updating your comments and questions. To emulate a low-accuracy MI module, we intentionally enlarge the training loss $l_s$ in Eq. 2 from 1.4e-4 to 3.0e-3. The accuracy 73.83% (as well as 83.08% accuracy of the original MI without $l_s$ enlargement) is measured on a test dataset against the victim model, which consists of 10,000 true training samples and 10,000 non-training samples.
> > >
> > > We are aware of the MI defense schemes in the literature, such as [NRA2018] and [SA2021]. Since we cannot enumerate them all in the experiment and we would like to know if MExMI is future proof, we emulate these defenses by raising the training loss of shadow model, as most existing defenses eventually reduce the accuracy of shadow models and thus that of MI attacks.
> > >
> > > _Reference_
> > >
> > > _[NRA2018] Nasr, Milad, Reza Shokri, and Amir Houmansadr. "Machine learning with membership privacy using adversarial regularization." Proceedings of the 2018 ACM SIGSAC conference on computer and communications security. 2018._
> > >
> > > _[SA2021] Shejwalkar, Virat, and Amir Houmansadr. "Membership privacy for machine learning models through knowledge transfer." Proceedings of the AAAI Conference on Artificial Intelligence. Vol. 35. No. 11. 2021._

---

### Official Review · Reviewer_cjTV · 2022-07-11

**Rating:** 7
**Confidence:** 4
**Soundness:** 3 good
**Presentation:** 3 good
**Contribution:** 3 good

**Summary:**

In this paper, the authors show that ME and MI can reinforce
each other through a chained and iterative reaction, which can significantly boost
ME attack accuracy and improve MI by saving the query cost.Al

**Questions:**

Why are there so many references?

**Strengths And Weaknesses:**

Pros:

Paper is well organized

Paper is written in Good English


Cons:

Related work should be moved to the beginning

Methodology and algorithmic content is missing

Text in Figures is small (graphs)

---

> ### Author Response · Authors · 2022-08-02
> **Author Response to the Questions of Reviewer cjTV**
>
> **1, Related works should be moved to the beginning.**
>
> In Section 1, we have briefly described essential related works as background knowledge. Since the section “Related Works” is more comprehensive and covers those topics that are less relevant to our work, it is placed at the end so as not to distract the readers.
>
> **2, About the “missing sections” Methodology and Algorithmic**
>
> Some algorithmic contents are in the supplementary material (Appendix) in the interest of space.
>
> **3, About the text size in figures**
>
> We shall increase the legend font size in the figures in the final version.
>
> **4, Why are there so many references?**
>
> Some of the references occur in the supplementary material (Appendix), not in the main paper.

---

> > ### Author Response · Authors · 2022-08-08
> > **Dear Reviewer, have we addressed your concerns?**
> >
> > Thank you for your support of this work. In case you've got some extra time, there are a few discussions between us and the other reviewers. If you have any questions, we would be happy to answer them within the reviewer-author discussion period.

---

### Official Review · Reviewer_N4YB · 2022-07-11

**Rating:** 7
**Confidence:** 3
**Soundness:** 3 good
**Presentation:** 2 fair
**Contribution:** 3 good

**Summary:**

This paper proposes a new framework that combines model extraction (ME) and membership inference attack methods such that, both attacks can mutually reinforce each other in the attack process, and eventually lead to efficient and effective ME and MI attacks. The experimental results show that, using the combined approach, the fidelity of the attack can be enhanced greatly compared to baseline ME attacks, while the MI in the mean time maintain relatively comparable performance compared to existing MI attacks that uses much more queries.

**Questions:**

The weakness part in the review lists up the requested changes and clarifications. An additional question for the authors is: (ME and MI attacks are not in my area) are the considered baselines in this paper reflect the state-of-the-art? It seems the adopted ones in this paper are rather "old", considering how hot the area is.

**Limitations:**

The limitations of the paper and the possible ways of improvements are outlined in the weakness section.

**Strengths And Weaknesses:**

The idea of combining ME and MI attack seems to be novel and interesting, and the empirical results also mostly support the claims made in the paper. Overall, I like the originality of the idea presented in this paper and the presentation is also clear. However, I also have the following concerns:
1) it is better to explain how much of a fidelity improvement is considered significant in practical applications. For example, in Table 1, the author listed the different fidelity scores when the level of training data homogeneity varies, and to me, all the fidelity scores are very high and the improvements may not be that significant. Explaining this will also help readers to better understand significance of the results.
2) I am not sure how accurate it is claim that training shadow MI model [40] requires training data of same size as the victim's training set. Their method is definitely applicable even if the shadow model training data only contains subset of the victim's training data. Therefore, this part should be stated clearly.
3) for MI attack results, it is extremely difficulty to properly compare the baseline MI attack [38] and the MExMI approach. The term "on par" is vague especially when we use "precision" and "recall" to compare them individually. Using some unified metrics such as F1-score, AUC scores or simply the MI accuracy might give us better pictures. Note that, MexMI does not have to outperform existing MI attacks that require significantly more resources.
4) the baseline methods for both ME and MI attacks might not be sufficient. For example, the MI attack [40] is also not included in the paper for baseline comparisons. In general, I believe the authors can first compare MExMI against various existing baselines and then do an ablation study on the impact of each module in MExMI.

---

> ### Author Response · Authors · 2022-08-02
> **Author Response to the Questions of Reviewer N4YB**
>
> **1, How much of a fidelity improvement is considered significant?**
>
> In the Introduction, we mention that high-fidelity model extraction attacks can provide a substitute model for follow-up attacks such as adversarial samples, model evasion, etc. Then in Appendix F.3, we evaluate the impact of fidelity on adversarial sample attacks. As the results show, if the copy model’s fidelity increases from 84.65% to 92.41% by MExMI, the success rate of adversarial sample attacks will increase from 51.76% to 63.87%. This shows that fidelity improvement can have amplification effect on other attacks, so a minor improvement on fidelity can still be significant in terms of potential threats.
>
> **2, How accurate is it to claim that the training shadow membership inference (MI) model [40] requires training data of the same size as the victim's training set?**
>
> This claim is correct, because the original claim in [40] has two occurrences: (1) "The training datasets of the target and shadow models have the same format but are disjoint." and (2) "The training set and the test set of each target and shadow model have the same size, and are disjoint.", both of which is consistent with this claim.
>
> **3, Use unified metrics for membership inference (MI) attack.**
>
> We have added metric “MI accuracy” in the revised paper to evaluate the MI performance, as in [38] [28]. The results show that in terms of MI accuracy, the performance of MI in MExMI can be described as “on par with” ML-Leaks [38].
>
> **4, More baselines of model extraction (ME) and membership inference (MI)**
>
> Thanks for your suggestions. We have added more state-of-the-art model extraction attacks as baselines in Section 5.3.2 Table 2 of the revised paper, including PRADA [23] and DFME [44]. Knockoff [32] is excluded from the experiment due to its strong assumptions about the pool’s coarse-to-fine hierarchy (e.g., the top-down nodes of a tree is animal, bird, and sparrow) for reinforcement learning. In terms of query efficiency, the results show that MExMI has a significant advantage over all baselines. Specifically, in the image classification experiments, with a 16K query budget, the fidelity of DFME is 11.20%, and PRADA is 61.32%, while the MExMI is 94.07%.
>
> As for MI baselines, since the MI module in MExMI is changeable (e.g., we can use either metric-based shadow-model MI or unsupervised MI), we only compare with the original MI methods in [38] and [40] due to space limitation. Nonetheless, better MI baselines can only improve the MExMI performance, not worsen it.
>
> **5, The adopted model extraction (ME) and membership inference (MI) works are too ‘old’?**
>
> Although there are newer ME attacks as reviewed in “Related Work” section, the adopted ME and MI works are state-of-the-art in its own category. For example, ActiveThief [33] is still the best known pool-based active model extraction (the same category as MExMI), in terms of query efficiency. In Table 2 of Section 5.3.2, we do try newer attacks such as DFME [44], but they are outperformed by ActiveThief, mainly because these new ME attacks (including MAZE [24]) are in another category called “synthesizing model extraction”. To build an accurate synthesizing model, these attacks must consume vast number of queries to warmup, which is already beyond the totally query budget in a practical ME attack, e.g., tens of thousands of queries.
>
> As for MI works, as the MI module in MExMI is changeable (e.g., we can use either metric-based shadow-model MI or unsupervised MI), we only compare with the original MI methods in [38] and [40] due to space limitation. Nonetheless, better MI works can only improve the MExMI performance, not worsen it.

---

> > ### Comment · Reviewer_N4YB · 2022-08-07
> > **Concerns are addressed**
> >
> > Thanks for the clarification, my raised concerns are well addressed and I am adjusting my score accordingly.

---

> > > ### Author Response · Authors · 2022-08-08
> > > **Thank You for Updating Your Review**
> > >
> > > We are glad to hear that we have addressed your concerns. We appreciate the updating score and the thorough review.

---

### Meta-Review · Area_Chair_fLqB · 2022-08-23

**Recommendation:** Accept
**Confidence:** Less certain

**Metareview:**

This paper studies the problem of securing a model once it is published as a service. In most prior studies the focus was on either protecting the model from extraction attacks (ME) or from identifying the data used for training the model (MI). The authors propose that a simultaneous attack on both surfaces is even more powerful since the MI attack provides more information to the ME attack.

The reviewers found that this work is interesting, and results are relevant to the committee as they can highlight the need for protecting the multiple surfaces of attack against ML models. However, there is a concern about the results presented in response to the comments of reviewer LJzA: in the response, the authors reported on the result of an experiment in which a defense against MI was simulated. From the gist of the paper, one would expect that this will result in some level of protection against ME attacks too. However, the results provide only weak support to this assertion. Therefore, we think the paper should address this inconsistency.


**Award:**

No

---

### Decision · Program_Chairs · 2022-09-14

Accept